# AI-Enabled Vessels Segmentation Model for Real-Time Laparoscopic Ultrasound Imaging

Ignas Kupcikevicius*[1, 4], Luca Boretto[1], Inger Annett Grünbeck[1, 4], Rahul Prasanna Kumar[1], Varatharajan Nainamalai[1], Seyed Mohammadmehdi Sadat Akhavi[1, 3], Bjørn Edwin[1, 2, 3], and Ole Jakob Elle[1, 4]

[1]The Intervention Center, Rikshospitalet, Oslo University Hospital, Oslo, Norway
[2]Department of Hepato-Pancreatic-Biliary Surgery, Oslo University Hospital, Oslo, Norway
[3]Institute of Clinical Medicine, Faculty of Medicine, University of Oslo, Oslo, Norway
[4]Department of Informatics, University of Oslo, Oslo, Norway

## Abstract

Laparoscopic ultrasound (LUS) is essential for assessing the liver during laparoscopic liver resections. However, the interpretation of LUS images presents significant challenges due to the steep learning curve and image noise. In this study, we propose an enhanced U-Net-based neural network with a ResNet18 backbone specifically designed for real-time liver vessel segmentation of 2D LUS images. Our approach incorporates five preprocessing steps aimed at maximizing the training information extracted from the ultrasound sonogram region. The modified U-Net model achieved a Dice coefficient of 0.879, demonstrating real-time performance at 40 frames per second and enabling the development of advanced ultrasound-based surgical navigation solutions.

## 1 Introduction

Liver cancer remains one of the top 10 deadliest cancers worldwide, resulting in approximately 750,000 annual deaths [1]. The reason for its mortality rate is late diagnoses, limited treatment options, and underlying liver disease with aggressive tumor biology [2]. To locate liver tumors and vessels during laparoscopic liver surgery, clinicians are using the laparoscopic ultrasound (LUS), which helps to navigate and to avoid unnecessary damage during liver resection or ablation. LUS is a radiation-free medical device, portable and cost-effective. It provides real-time images by capturing ultrasound reflected pulses from soft tissues and bones [3]. All of these LUS benefits give clinicians the ability to effectively diagnose liver cancer, such as hepatocellular carcinoma and other metastases. Additionally, LUS allows visualization of essential liver structures, the portal vein, hepatic veins, and bile ducts.

While valuable, LUS comes with several drawbacks. A key problem is speckle noise, an artifact from ultrasound waves, that interferes as reflected off tissue microstructures. This effect lowers overall image quality [4]. Vessel boundaries can also appear unclear because of differences in tissue echogenicity - the way tissues reflect sound. When boundaries fade, tracking blood vessels during surgery becomes more difficult. Finally, underlying conditions like fatty liver or cirrhosis are causing liver texture changes which interfere with the interpretation of ultrasound scans [5].

These imaging issues limit how effectively LUS can guide surgeons during liver procedures [6]. Several conventional techniques could be used to account for these challenges. One of the default modes of current ultrasound (US) systems, is Color Doppler mode, which can be used to visualize blood flow by detecting frequency shifts in moving blood cells and to enable real-time assessment of vascularity. However, it has a relatively small region of interest, and its effectiveness is heavily dependent on the operators' skill, which might introduce inconsistency in the interpretation of the LUS data [7].

Another traditional visualization method is a Contrast-Enhanced Ultrasound (CEUS). It uses microbubble contrast agents to improve the visibility of blood vessels. Unfortunately, this method requires careful timing to capture the best blood flow enhancement after the contrast is given, which can be difficult in busy surgical environment [8]. Traditional segmentation algorithms, such as region growing, thresholding, and clustering, have also been employed for tissue segmentation [9]. All of them require manual tuning of thresholds value and seed points, which limits their robustness in handling the complex and heterogeneous tissue structures present in ultrasound images.

### 1.1 Related work

Over recent years, deep learning has become a common approach for automated vessel segmentation. Reported performance from different studies varies widely across imaging modalities, with Dice scores of 0.734 for ultrasound [10], 0.928 for MRI [11], and 0.814 for CT [12]. Dice scores come from different datasets and are not directly comparable, however, they show that ultrasound remains a challenging

---

*Corresponding Author.

Proceedings of the 7th Northern Lights Deep Learning Conference (NLDL), PMLR 307, 2026.

modality for vessel segmentation. U-Net-based architectures, have been recognized as the gold standard for semantic segmentation tasks [13]. Their encoder-decoder structure and skip connections have made them adaptable to enhancements such as adding residual blocks (ResU-Net) [14], dense connections (DenseU-Net) [6], attention gates (Attention U-Net) [15], or transformers (TransU-Net) [16]. Although these studies have demonstrated competitive results in segmenting various biological tissues from ultrasound data, only a few have explored the performance of real-time segmentation [6, 17].

Real-time ultrasound image segmentation is a complex task due to the noise and inconsistent data. Preprocessing is often employed to suppress speckle noise and reduce artifacts, but this adds computational overhead. Varying echogenicity makes boundary detection difficult. This means the model requires careful tuning to remain reliable under these imaging conditions. Post-processing techniques, such as mask refinement for frame-to-frame consistency, or resizing output to the original resolution, add further computational load. This makes it difficult to balance between models tuned for accuracy and models tuned for the speed required in real-time applications. Smistad et al. [18] used an Artificial Intelligence (AI) model to segment blood vessels, nerves, and bone structures during anesthesia-related procedures, and showed a promising real-time performance. However, the predictions were made on a frame-by-frame basis without considering the temporal information in the sequential ultrasound data, which could have inherent potential information to enhance the performance.

## 1.2 Contribution summary

This paper presents an automated AI-enabled LUS model for real-time vessel segmentation, developed to support liver surgery and improve intraoperative guidance. Our main contribution is an end-to-end, real-time workflow that automatically extracts and masks the ultrasound sonogram, applies CLAHE tuned to our LUS data, and uses a triplet-frame input with a lightweight ResNet18 backbone. Together, these components improve vessel detection and segmentation continuity while keeping inference speed compatible with intraoperative use. Below, we outline the methodological and dataset contributions that form this clinically-oriented pipeline.

1. Fully anonymized LUS liver video data was locally acquired and annotated with the assistance of experienced clinicians, and all annotations were verified and approved by a radiologist.

2. A dynamic approach was developed to extract the ultrasound sonogram from video frames. It enabled precise masking of the imaging area and

prevented the network from learning irrelevant background features, thereby improving segmentation accuracy without compromising real-time performance.

3. The triplet input setup, similar to ones used for LUS-CT co-registration [19] and for object recognition [20], was integrated into a lightweight ResNet18 U-Net model, enhancing segmentation quality by introducing contextual information between frames.

4. Contrast Limited Adaptive Histogram Equalization (CLAHE) [21] was applied and optimized for our dataset. It enhanced vessel boundaries and improved lumen visibility, which resulted in increased segmentation accuracy.

5. A comprehensive study was conducted to evaluate the performance of different U-Net family encoders, focusing on both segmentation accuracy and real-time inference efficiency.

The AI-generated 2D liver vessel segmentation masks can also be used for 3D vessel reconstruction, aiding in image registration between preoperative and intraoperative stages.

## 2 Proposed methodology

The proposed method uses an encoder–decoder architecture with a ResNet18 backbone to perform real-time vessel segmentation in laparoscopic ultrasound frames.

### 2.1 Pre-processing pipeline

As shown on the left side of Figure 1, all the LUS frames undergo standardization before entering the network to: a) emphasize learning focused on the acoustic sonogram, b) stabilize contrast across different cases, acquisition depths, and sonogram shapes, and c) ensure a consistent input shape while preserving spatial geometry. The specific preprocessing steps (1 to 5) visualized in Figure 1 will be described in detail as follows:

1. **Grayscale conversion**: Our recorded LUS videos contain identical R, G and B channels (stacked intensity), therefore, we converted frames to a single grayscale channel to remove redundancy, reduce computation and memory use, and avoid learning artificial color patterns that are not part of the actual ultrasound signal.

2. **Sonogram detection:** To locate the US sonogram as a trapezoid in frame coordinates, the dynamic contour-based detection with a fallback heuristic was developed and used. For the fallback, the most recently detected good coordinates

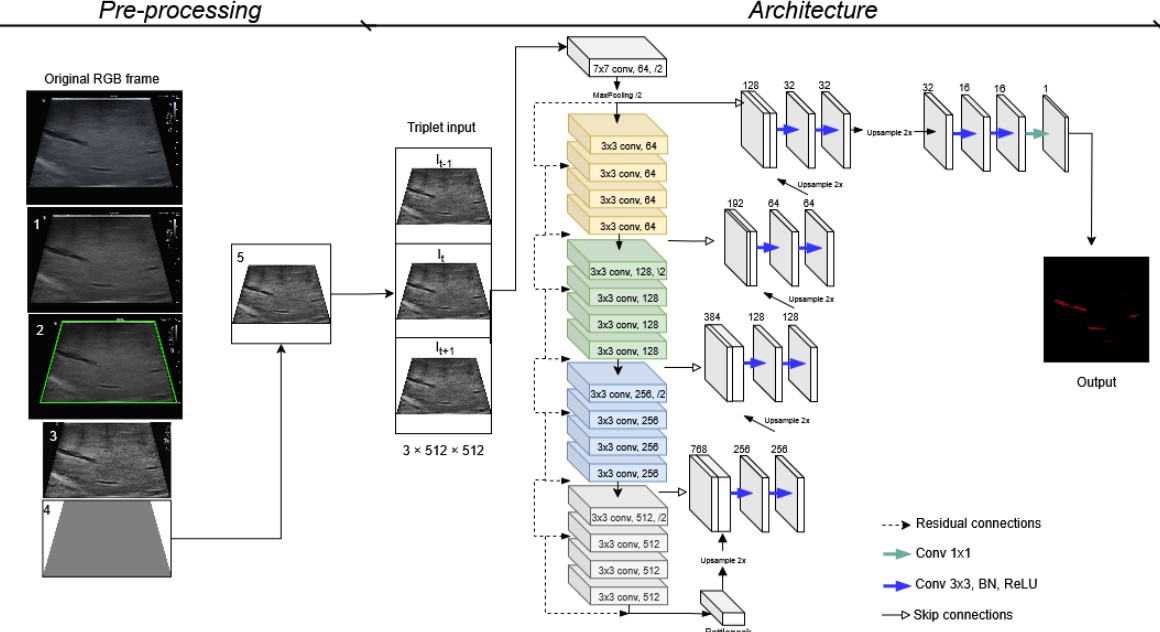

**Figure 1.** Overview of the proposed vessel segmentation pipeline. The raw RGB laparoscopic ultrasound frames undergo five preprocessing steps, are arranged into temporal triplets, encoded with a ResNet18 backbone, and decoded with a U-Net decoder to generate the final binary mask of the vessels.

were used. The trapezoid is defined by four corner points, stored in a $4 \times 2$ matrix $P^{\mathrm{trap}} \in \mathbb{Z}_{\geq 0}^{4 \times 2}$, where each row holds the $(x, y)$ coordinates of one corner in a set of intigers $\mathbb{Z}$. By using this locally developed method, we managed to successfully detect the sonogram contour even when the acquisition depth was changing.

3. **Tight crop and CLAHE:** After locating the sonogram coordinates, we applied a tight rectangular crop to remove unnecessary background. Following the CLAHE application strategy of Ansari et al. [6], we tuned two hyperparameters - the contrast limiting threshold (clipLimit) and the grid size for histogram equalization (GridSize), using a small-scale grid search on a validation subset. The selected values, clipLimit = 2.0 and GridSize = $8 \times 8$, consistently improved image contrast and produced better segmentation performance both quantitatively and qualitatively.

4. **Sonogram masking:** After cropping, the sonogram was isolated using a binary polygon mask (sonogram = 0, outside = 255). Whitening the background region prevents the model from learning irrelevant background patterns and reduces false positives outside the imaging area, ensuring the network focuses on anatomical features inside the sonogram. The original video frames contained non-anatomical elements such as text overlays, depth and distance markers, and interface graphics from the ultrasound device. Including these elements during training could lead the

network to associate them with anatomical structures, so removing them ensured that only clinically relevant image content was used. Similar masking-based extraction approaches are commonly applied in ultrasound preprocessing [22].

5. **Resizing and padding:** In the final step, we resized the ultrasound frame while preserving its aspect ratio to avoid geometric distortion of vessel structures, which is a standard operation in ultrasound imaging workflows [22]. After resizing, the image was symmetrically padded to 512 x 512 using bilinear interpolation to provide a fixed square input compatible with a wide range of network backbones.

By applying these five preprocessing steps to each LUS video frame, we direct the model's focus toward the sonogram area that contains vascular anatomy. CLAHE enhances local contrast for thin, low-contrast vessels, while aspect-ratio-based resizing prevents geometric distortions of tubular structures.

## 2.2 Network architecture

For this study, we tested and adopted a U-Net-based encoder–decoder network with multiple backbones from the U-Net family, including lightweight ResNet18 [23], MobileNet_v2 [24], DenseNet-121 [25], medium-sized ResNet50 [23], and Vanilla U-Net [13], as well as a larger model, InceptionNetV2 [26]. All encoders except the vanilla U-Net were used via the Segmentation Models PyTorch library [27],

which provides standardized open source implementations of these architectures. The selected encoder, ResNet18, and the standard U-Net decoder, used for training and inference, are shown on the right side of Figure 1. ResNet18, pretrained on ImageNet [28], provides residual blocks that support stable training on small medical datasets. To exploit temporal coherence, sequential frames are processed as triplets rather than individually, enabling motion-aware and more consistent predictions. In general, U-Net architecture was selected due to its strong performance in segmentation tasks and its skip connections which help retain fine spatial details that are often lost during the downsampling process.

### 2.2.1 Modified input layer: temporal triplet setup

Unlike conventional U-Net inputs that use a single 2D frame, we implemented triplets of consecutive frames to utilize the temporal information of sequential ultrasound video data. During training process, we used a symmetric triplet setup $[I_{t-1}, I_t, I_{t+1}]$, with the ground truth mask corresponding to the middle image $I_t$. This configuration allows the model to learn context from both past and future frames, possibly enhancing vessel continuity and robustness to speckle noise. During inference, since future frames are not available, we switched to a more standard triplet setup: $[I_{t-2}, I_{t-1}, I_t]$, which maintains the benefits of temporal context without compromising real-time performance.

### 2.2.2 Loss, optimization and training controls

In preliminary experiments with a Vanilla U-Net, we evaluated several segmentation loss functions commonly used in medical imaging. Dice loss [29], Focal loss [30], and Binary Cross Entropy (BCE) loss [31] functions were tested and compared. By looking at the Dice score curves in Figure 2 we noticed, that BCE was more stable and it reached slightly higher Dice values than Dice or Focal loss. Dice and Focal loss showed more fluctuations, suggesting less reliable optimisation. BCE was therefore selected as the primary training loss for all encoder backbones, as it offered the most consistent generalization and improvements. The Dice coefficient was used as the primary evaluation metric during model validation. The BCE loss is defined as follows:

$$\mathrm{BCE}(p, y) = -\left(y \log(p) + (1 - y) \log(1 - p)\right), \quad (1)$$

where $y \in \{0, 1\}$ is the ground truth, and $p \in [0, 1]$ is the predicted probability (after sigmoid function). We employed Binary Cross-Entropy with logits loss (*BCEWithLogitsLoss* in PyTorch [32]), which is equivalent to applying a sigmoid activation followed by binary cross-entropy, but implemented in a numerically more stable form.

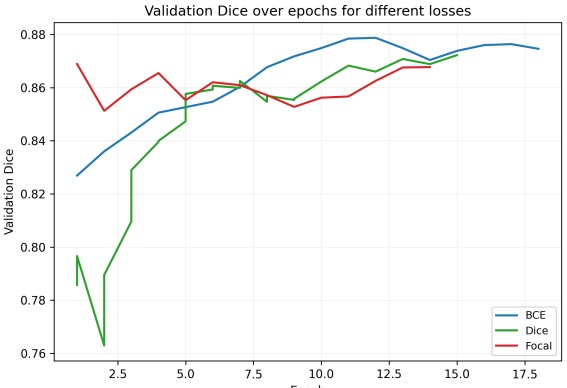

**Figure 2.** Validation Dice across epochs for three different loss functions (BCE, Dice, and Focal), using early stopping. These curves are smoothed, so the last few points may appear to increase slightly even though the underlying validation Dice already plateaued when early stopping triggered.

### 2.2.3 Optimization

All our networks were trained for up to 100 epochs under identical conditions to ensure a fair comparison across backbones. For optimization, we used Adam optimizer with a learning rate of $1 \times 10^{-3}$ and a learning rate scheduler (factor = 0.5, patience = 5) to improve convergence once the validation loss plateaued. During preliminary experiments, we evaluated several geometric and intensity-based augmentation strategies. Different encoder architectures responded inconsistently to these augmentations, with some showing improved performance and others degrading under the similar augmentation setup. To avoid introducing architecture-dependent bias and to maintain a controlled comparison across models, we therefore trained all networks without additional augmentation. To accelerate training and reduce GPU memory usage, automatic mixed precision was used. We used early stopping function to interrupt training after 10 epochs without improvement in the validation Dice score. The batch size was set to 18, determined by the available GPU memory (RTX 4080, 12 GB VRAM). Each channel of the triplet input was normalized to $[-1, 1]$ to improve training stability.

### 2.3 Dataset and data split

The dataset consists of laparoscopic ultrasound videos from 11 acquisitions, obtained from 9 patients. One patient contributed three acquisitions from separate sessions, each capturing distinct liver views, as confirmed by a radiologist, and treated as independent cases. By doing this we preserved

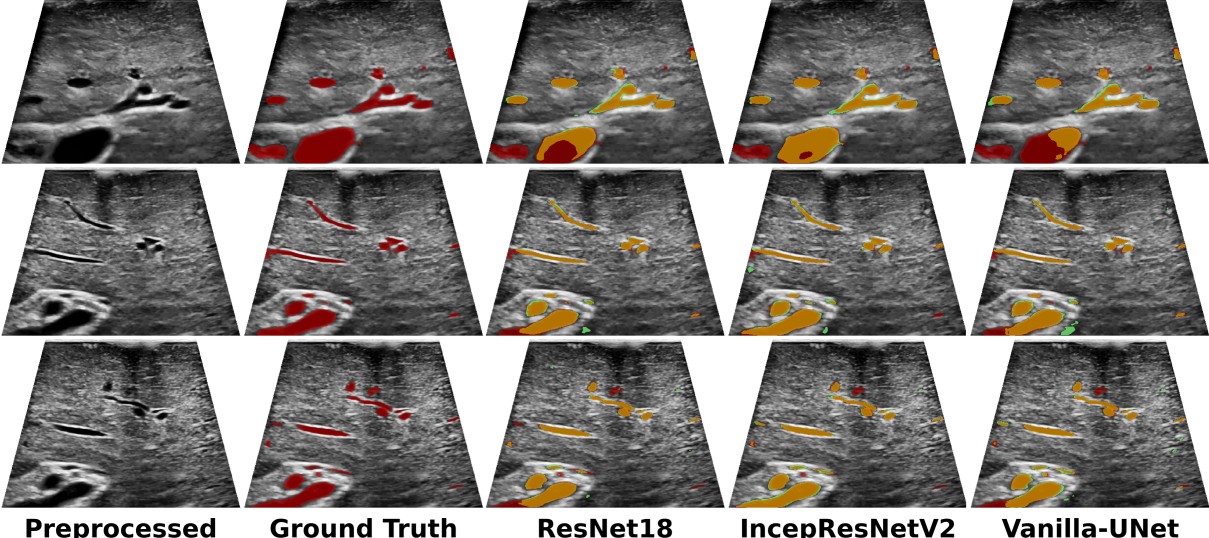

| **Preprocessed** | **Ground Truth** | **ResNet18** | **IncepResNetV2** | **Vanilla-UNet** |

**Figure 3.** Vessel segmentation results using three models.
Red: ground truth, Yellow: model predictions, Green: over-predicted regions.

.

all usable data and maximized the full size of the training set. All data were recorded using a commercial laparoscopic ultrasound system, and only video-format data were available. Access to raw ultrasound signals was not possible. From these videos, 2,200 sequential frames containing vessels were extracted, and pixel-wise binary masks were created and subsequently verified by a radiologist. To preserve independence between development and evaluation, one case was set aside as the test set. The remaining ten acquisitions were used for model development under a 5-fold cross-validation scheme, with eight used for training and two for validation in each fold. After cross-validation, a final model was trained on the same ten acquisitions using a 90/10 split to maximize the training sample while retaining an internal validation subset. Final performance was evaluated on the held-out test set.

## 3   Results and discussion

Various backbone architectures were tuned and compared within the proposed U-Net framework. Performance was evaluated using the cross validation protocol described in Section 2.3 to identify the most promising encoders. A final model was then trained and assessed on the held out test set to measure generalization on unseen data.

### 3.1   Quantitative results

Comparison results presented in Table 1 are from 5-fold cross-validation experiments, made with six backbone configurations. All tested encoders achieved Dice scores above 0.9 in 5-fold CV, demonstrating that vessel segmentation in LUS is a learn-able task. The consistent results across architectures indicate robustness and suggest that encoder selection can be made not only by the accuracy alone, but also by efficiency and deployment factors. When looking at the results from the test set, both InceptionResNetV2 and ResNet18 performed well, with InceptionResNetV2 reaching the highest Dice score of 0.888 and recall of 0.867. Deep architecture and Inception modules allow InceptionResNetV2 to capture multi-scale features, while residual connections help stabilize training similarly as in ResNet18. However, with a large number of parameters, the model is computationally heavy, leading to longer training times and slower inference compared to lighter encoders.

Despite being the lightest model, ResNet18 produced competitive Dice of 0.879 and high precision score of 0.923, benefiting from residual connections that support efficient gradient flow and stable feature learning. Additionally, the smaller parameter count noticeably improved the inference speed, preserving real-time capability even with our additional preprocessing steps. This balance of accuracy and efficiency makes ResNet18 particularly suited for laparoscopic ultrasound vessel segmentation, where reliable performance must be achieved under strict computational constraints.

### 3.2   Qualitative results

Selected segmentation examples from three test frames, shown in Figure 3, compare ground truth annotations with predictions from the lightweight ResNet18, the heavier InceptionResNetV2, and the medium-sized Vanilla U-Net. Across the models, there is a noticeable tendency for slight over-

**Table 1.** Comparison of vessel segmentation performance of different models: Dice coefficient (DC) with standard deviation (Std) from 5-fold cross-validation (CV), and DC, recall, precision, and Intersection over Union (IoU) on the test set.

| Encoder Type | 5-Fold CV | Test set | | | |
|---|---|---|---|---|---|
| | DC ± Std | DC | Recall | Precision | IoU |
| ResNet18 | 0.916 ± 0.002 | 0.879 | 0.840 | **0.923** | 0.783 |
| ResNet50 | 0.904 ± 0.002 | 0.856 | 0.811 | 0.912 | 0.748 |
| MobileNet_v2 | 0.906 ± 0.002 | 0.862 | 0.819 | 0.917 | 0.757 |
| Vanilla U-Net | 0.918 ± 0.003 | 0.859 | 0.827 | 0.897 | 0.753 |
| DenseNet121 | 0.901 ± 0.003 | 0.849 | 0.798 | 0.913 | 0.738 |
| IncResNetV2 | 0.926 ± 0.002 | **0.888** | **0.867** | 0.911 | **0.798** |

**Table 2.** Comparison of single-frame and triplet input ResNet18 U-Net models. Mean ± standard deviation is reported for all metrics. Arrows indicate whether higher (↑) or lower (↓) values are better.

| Metric | ResNet18 U-Net (Single) | ResNet18 U-Net (Triplet) |
|---|---|---|
| Per-frame DC vs. GT (↑) | 0.875 ± 0.031 | **0.879 ± 0.030** |
| Temporal DC (↑) | 0.932 ± 0.056 | **0.938 ± 0.045** |
| Temporal IoU (↑) | 0.877 ± 0.091 | **0.886 ± 0.073** |
| Flip rate (↓) | 0.008 ± 0.008 | **0.007 ± 0.007** |
| Boundary jitter (px, ↓) | 1.334 ± 1.257 | **0.980 ± 0.990** |

segmentation. While InceptionResNetV2 produced the most visually refined results, its computational complexity makes it less suitable for real-time deployment. In contrast, ResNet18 provided visually comparable masks, with segmentation quality not significantly inferior to that of Vanilla U-Net, despite being much lighter.

### 3.3 Impact of the triplet setup

To assess whether the proposed triplet input improves temporal stability compared to single-frame predictions, we trained a ResNet18 model with both single-frame and triplet inputs under otherwise identical settings. We then defined temporal consistency metrics, following recent studies [33, 34], and reported them in Table 2. *Temporal Dice* is defined as the Dice coefficient between consecutive frame predictions, averaged over the sequence, while *Temporal IoU* is defined analogously using the IoU.

During training, the model predicts the segmentation of the central frame in each temporal triplet, which is a common setup allowing the network to learn temporal context from both future and past frames. During real-time inference, the future frame is not available, and the model therefore operates on the current frame and its two earlier frames. This creates a small mismatch between training and inference conditions and remains a limitation of the present implementation. A future extension could use only past frames during training to fully align with real-time constraints.

In addition, we evaluated prediction stability across time. Following Rebol et al. [35], we measured *Flip-rate,* the proportion of pixels whose labels switch between consecutive frames (e.g., a vessel pixel that disappears and reappears). This captures segmentation "flickering" over time. However, in our experiments, Flip-rate values were consistently close to zero, likely reflecting both the strong class imbalance (vessel vs. background) and the generally high segmentation accuracy. Thus, the Flip-rate confirms the absence of large temporal instabilities in both compared models.

To quantify boundary stability, we used the idea from Perazzi et al. [36]. We define *Boundary jitter*

as the average displacement (in pixels) of the predicted vessel boundary between consecutive frames, capturing small shifts of vessel contours. The results indicate that our model performs better when trained with adjacent frames, enhancing all temporal metrics. We also assessed the per-frame Dice score against the ground truth (GT), which showed slight improvement. Additionally, the triplet input configuration improved temporal consistency. It produced masks that were more stable and less prone to flickering compared to the single-frame predictions shown in Figure 4. The triplet-frame setup also reduced both false positive and false negative predictions. Overall, it generated smoother masks and a more stable stream during real-time experiments.

Beyond the single-frame versus triplet comparison, we also conducted a follow-up experiment to examine how different temporal input configurations influence model behavior. We compared single-frame, triplet, five-frame, seven-frame, and a far-frames (frames at positions: −3, 0, +3), all under identical training conditions. All temporal configurations performed better than the single-frame baseline in terms of training convergence and validation accuracy, confirming that temporal context is beneficial for LUS vessel segmentation. Full training curves for this ablation are provided in Appendix A. Increasing the number of adjacent frames (five or seven) did not yield additional gains, suggesting that neighbouring frames are highly redundant. Interestingly, the far-frame setup, which introduces broader temporal separation, produced the most stable validation loss and the highest validation Dice. This suggests that temporally distant frames provide more useful information than tightly clustered ones. Such observation may be worth examining further to understand how different temporal sampling strategies affect model performance. Overall, the results support the use of temporal inputs and show that the triplet configuration offers a practical balance between accuracy, stability, and real-time feasibility. However, these trends are specific to our dataset and acquisition setup, and different probe motion dynamics or frame sampling rates may lead to different temporal dependencies.

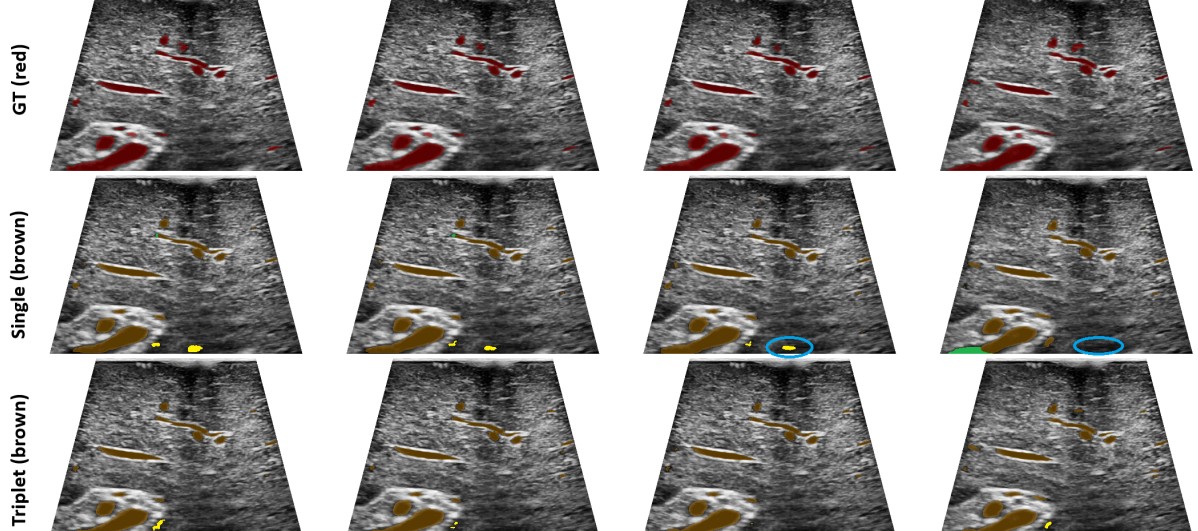

**Figure 4.** Qualitative comparison of vessel segmentation results from single-frame and triplet-based ResNet18 U-Net models. Green color highlights false negatives (under-predicted vessels), while yellow color marks false positives (over-predicted regions), and blue circles indicate temporal flipping across frames.

## 3.4 Real-time experiment and results

To test deployment under realistic conditions, the models were evaluated on an unseen laparoscopic ultrasound video using the full real-time pipeline, including preprocessing, frame-wise inference, and post-processing to mimic the clinician's view. The developed dynamic contour detector enabled the model to adapt to changing imaging depths without introducing artifacts, maintaining high frames per second (fps) even with intensive preprocessing and post-processing.

The real-time performance metrics of the tested models are summarized in Table 3. The final column reflects the results observed during continuous video playback in fps. Despite InceptionResNetV2 achieving the highest Dice score, it was the slowest for inference. The literature suggests that a frame rate of 30 fps is generally sufficient for real-time performance [37, 38]. A couple of tested models, including ResNet18, ResNet50, and MobileNet_v2 reached and sustained this fps target, with ResNet18 also allowing for potential further tuning if needed. Notably, this model delivered segmentation quality comparable to that of InceptionResNetV2 while achieving a stable throughput of 40 fps, making it the most viable candidate for intraoperative segmentation deployment.

## 3.5 Dataset limitations

The dataset used in this study was relatively small, consisting of 11 acquisitions and 2,200 images, however it is comparable to the dataset sizes reported in other liver vessel segmentation studies [6, 10]. We recognize, that limited number of acquisitions constrains the amount of independent data available for

**Table 3.** Comparison of model complexity and performance. Parameter count, forward-pass inference time, model size, and measured real-time fps (including preprocessing and postprocessing in the ultrasound pipeline) are reported.

| Encoder name | Parameters | Time (ms) | Model size (MB) | FPS |
| --- | --- | --- | --- | --- |
| ResNet18 | 14,328,209 | **5.48** | 56.08 | **40** |
| ResNet50 | 32,521,105 | 9.03 | 127.38 | 36 |
| MobileNet_v2 | 6,628,945 | 7.55 | 26.17 | 37 |
| Vanilla U-Net | 31,037,633 | 12.63 | 121.33 | 29 |
| DenseNet121 | 13,607,633 | 16.97 | 53.78 | 29 |
| IncResNetV2 | 62,029,297 | 29.21 | 243.04 | 21 |

validation. As we kept one test set for final testing, the remaining ten acquisitions were used for 5-fold CV and model development. Despite this limitation, the variation between folds in the 5-fold CV results, presented in Table 1, remained low. This indicated a stable model performance. In the future, other evaluation techniques such as 3-fold cross-validation or leave-one-out validation could be explored. Finally, if more annotated LUS data become available, the model can be retrained to further strengthen generalizability.

## 3.6 Model selection limitations

CNN-based backbones, used in this study, are not the newest architectures, but they remain widely applied and competitive in medical image segmentation. Other types, like transformer based segmentation models represent a more recent research direction, but they typically require much larger datasets and higher computational resources, which were not available in this study. Given our focus on developing an accurate and real-time vessel segmentation

model suitable for intraoperative use, we prioritized architectures that offer strong performance with limited data and efficient inference speed. For these reasons, transformer based methods were not further explored, but they remain a potential extension if more annotated data become available.

## 3.7 Dataset characteristics and clinical context

The dataset consists intraoperative ultrasound sequences collected from patients with various underlying liver conditions, including fatty liver and cirrhosis, which contribute to variability in image appearance. During the annotation process, the reviewing radiologist noted several cases showing typical features, including signs of fatty liver or cirrhotic change, as well as cysts, tumors, or marks from prior ablation procedures when visible within the 200 frame acquisitions. This variability reflects the real world conditions under which LUS vessel segmentation models must operate. While the dataset includes a wide spectrum of liver pathologies, the present study did not perform a pathology specific analysis of model performance. Such an investigation would be a valuable extension for future work.

## 3.8 Multimodal extension

We also considered whether the method could be extended to a multimodal setting by, for example, incorporating Doppler information, which highlights vascular structures through flow and velocity patterns. Prior work, such as Jiang et al. [39], has shown that combining Doppler with B-mode data can improve vascular segmentation by providing complementary physiological information. However, our retrospective LUS dataset contained only B-mode recordings and did not include any Doppler channels, making such multimodal fusion currently infeasible. Exploring Doppler-augmented segmentation therefore remains an interesting direction for future research.

## 4 Conclusion

In this paper, we presented a five-step preprocessing framework combined with a triplet-based ResNet18 U-Net model for real-time laparoscopic ultrasound image segmentation, achieving competitive Dice scores for liver vessel segmentation. Key contributions include a dynamic contour detector that improved generalization across varying depths and a triplet input setup that enhanced the temporal stability of vessel segmentation. We also evaluated real-time performance and mask quality, confirming the feasibility of deployment. Future work will focus on direct integration with live ultrasound streams and extension to 3D vessel reconstruction.

## Acknowledgments

The authors acknowledge the guidance and support of Rafael Palomar. This study was carried out as part of the EU project HoloSurge, project number 101137233, which aims to develop a multi-modal 3D holographic tool and real-time guidance system.

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

# Appendix A. Brief Analysis of Temporal Frame Configurations

To investigate the effect of different temporal input configurations, all models were trained on the same patient split, with Case 8 held out as the fixed test set and excluded from this experiment. The remaining subjects were used for training and validation, and early stopping was applied based on the validation macro Dice. Figures (A.1–A.3) summarize training loss, validation loss, and validation Dice across epochs for all temporal setups.

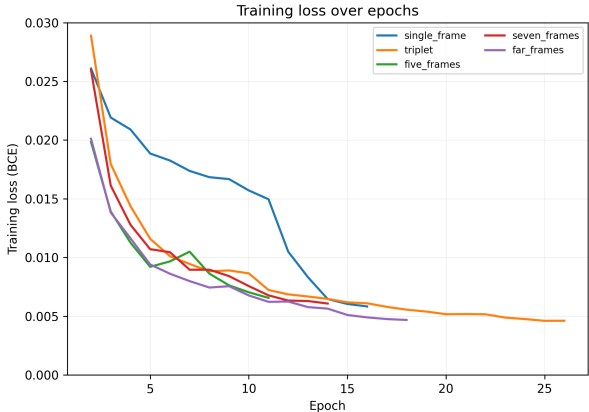

**Figure A.1.** Smoothened training loss over epochs for all temporal input configurations, using early-stopping.

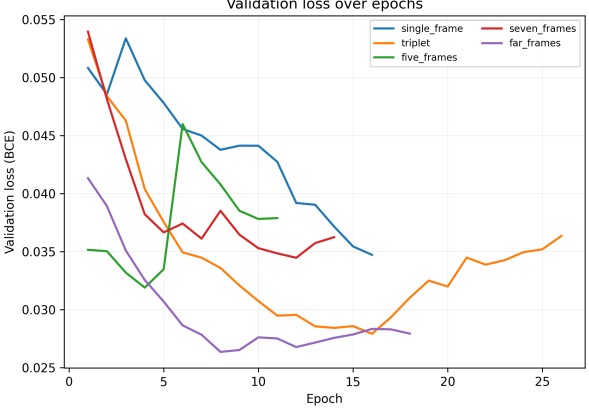

**Figure A.2.** Smoothened validation loss over epochs for all temporal input configurations, using early-stopping.

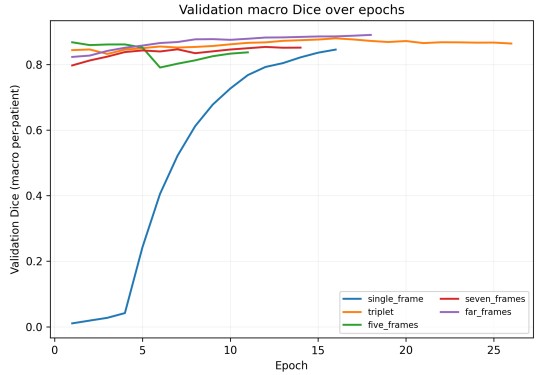

**Figure A.3.** Smoothened validation macro Dice across epochs for different temporal input configurations, using early-stopping.

