# OpenReview forum: "AI-Enabled Vessels Segmentation Model for Real-Time Laparoscopic Ultrasound Imaging"
_NLDL.org/2026/Conference — NLDL 2026 Spotlight_

### Official Review · Reviewer_sXEu · 2025-10-05
**Review of AI-Enabled Vessels Segmentation Model for Real-Time Laparoscopic Ultrasound Imaging**

**Rating:** 4
**Confidence:** 5

**Summary:**

The paper investigates methods for real-time segmentation of vessels in laparoscopic ultrasound imaging. The study is carried out on 11 cases with a total of 2200 image frames, where the vessels were annotated. Six backbone methods were investigated that all employed a U-Net framework. The IncResNetV2 obtained the highest segmentation accuracy, but since it is the largest model, it was also the slowest. The smaller ResNet18 achieved good segmentation accuracy while also having the fastest execution time; therefore, it is the recommended method.

**Strengths:**

Overlaying ultrasound images with vessel segmentation masks in real-time is an interesting problem, which is well motivated. Basing the investigation on a U-Net framework is a good choice, and the chosen six models represent different CNN architectures from lightweight to larger models. The study is based on 2200 ultrasound frames that have all been annotated, which is a significant effort. The experiments on segmentation accuracy and runtime are meaningful and thorough.

**Weaknesses:**

The study is carried out on a small dataset, and with the current description, it can be difficult to judge the validity of the findings. By addressing the detailed points below, it might be easier to judge the findings.

The six backbone methods used in the study are all relatively old, and none of them are cited in the paper. Were they implemented by the authors for this paper, or were they implementations from somewhere else? There are also no citations to the six backbone models. As far as I can tell, no transformer-based models or other modern architectures were included. Why not? They might be slow or not have enough data to train on, but including modern architectures would be interesting.

In the introduction, the study is motivated by fatty liver and cirrhosis, but this does not seem to be included in the dataset. Please provide more details on the data and the cases chosen.

Doppler will give movement direction, which is characteristic of blood vessels. Why does that not support the segmentation of the vessels?

Detailed comments.

In lines 77-80, segmentation accuracies are listed. But this is for other datasets, and these numbers are not meaningful in this context. The numbers are described as being impressive, but that depends on the context in which they are used.

In line 115, it is stated that the paper aims to develop an AI algorithm. I suggest rephrasing this to say that the model has been developed (it might not be in use).

In lines 122-123, it is stated that the annotations are under review. Does this mean that the annotations are not precise? I suggest that you rephrase this to tell how annotations were done or rerun experiments on reviewed data.

In lines 124-125, it is stated that the sonograms are extracted from video. Is the data recorded on commercial instruments with no access to the raw data? It would be good to have these details in the data description.

In lines 127-128, it says that the network should not learn irrelevant features. Can you tell more about what features it might start learning from?

In the section starting at line 236, the temporal triplets are explained. Can you explain why it is advantageous to train the model on the central frame when it has been used on the last frame at inference time? Why not both train and do inference on the last frame?

In lines 278-279, it is stated that no augmentations are employed. Why is that? You only have 2200 images, so data augmentation may be beneficial.

In line 282, it is claimed that mixed precision will not compromise accuracy. Was this tested? If not, this claim cannot be made.

In lines 282-285, it is stated that early stopping prevents overfitting. This is not correct. Choosing the model with the lowest validation loss prevents the model from overfitting. Early stopping just ensures that a potentially overfitted model is trained excessively long, and computation and time are wasted.

In lines 291-292, the data is described with 11 cases. Is that 11 patients? Please explain in more detail.

In lines 293-295, the frames are described. Were they all annotated? How is the distribution of the 11 cases?

In lines 299-300, the data split is described. Only training and validation are described. What about test data? Were the frames selected randomly? If they are randomly selected, neighbouring frames might be in the training and validation set. These images might appear very similar and therefore bias the performance, indicating unrealistically high performance. It might also give a wrong image of which model performs the best. I also suggest that the description of who did the annotation and how it was done be included here. Please explain how the data was treated.

In lines 307-308, a leave-one-case-out test set is described. Please explain that in more detail and more clearly. It might answer some of the questions above.

The colors chosen in Figures 2 and 3 for the segmentation overlay are not very clear. Especially, the wrongly segmented parts in Figure 2 are almost invisible. The blue rings help, but you could choose a brighter color. The brown color in Figure 3 is also difficult to see.

**Justification:**

The paper is an interesting study that is worth sharing. As seen from my list of weaknesses, I suggest many clarifications. But if this is added, I find the paper acceptable.

---

> ### Author Rebuttal · Authors · 2025-10-22
>
> To Reviewer sXEu
>
> Author’s Response:
> We sincerely thank the reviewer for highlighting the Strengths and Weaknesses.
>
> 1st weakness: We acknowledge the limitation of working with a small dataset and will carefully address all the detailed points raised below to better clarify and support the validity of our findings.
>
> 2nd weakness: The backbone encoders used in our study were implemented through the open-source SMP library, which provides standardized implementations of commonly used CNN architectures for semantic segmentation. The U-Net (vanilla) model was implemented separately following the original architecture described by Ronneberger et al. We will include the corresponding citations in the revised manuscript.
>
> CNN-based backbones are not the newest architectures, but they still remain widely adopted and competitive in medical image segmentation. The transformer-based segmentation models represent a more recent research direction, but they require much larger datasets and higher computational resources. Given our focus on developing an accurate and real-time vessel segmentation model suitable for intraoperative use, we prioritized architectures that offer strong performance with limited data and efficient inference speed. For these reasons, transformer-based methods were not further explored in this study.
>
> 3rd weakness: In the introduction, we mentioned that underlying liver conditions such as fatty liver and cirrhosis changes tissue texture and make image interpretation more challenging for clinicians. It is related to our dataset, as all cases were collected from patients with underlying liver pathology. During the validation of dataset annotations, the radiologist provided few remarks indicating that some cases showed specific pathological features (e.g., signs of fatty liver or cirrhotic changes, cysts, tumors and marks of previous ablations) when visible within the 200-frame acquisitions. However, a detailed analysis of how each underlying condition affected the model’s performance was outside the scope. We agree that such an investigation would be valuable for future work.
>
> 4th weakness: We agree that Doppler information would support vessel segmentation, as it provides flow and velocity characteristics that are specific to blood vessels. Doppler data can indeed be combined with B-mode ultrasound, for example by stacking modalities channel-wise, to incorporate spatiotemporal context and improve segmentation accuracy as done by Jiang et al. Unfortunately, our dataset did not include Doppler recordings, making this integration infeasible in the current work. We agree that extending the method to multimodal data would be an interesting direction for future research.
>
> Author’s Response to detailed comments:
>
> 1. We agree that the reported Dice scores come from separate datasets and cannot be directly compared. Our goal was not to make a quantitative comparison but to show that segmentation performance in CT and MRI tends to be a bit higher. Ultrasound segmentation remains a less explored task, which motivates the relevance of our study.
> If accepted, we will revise the phrasing in the introduction to clarify the purpose and context of these numbers.
>
> 2. We agree that statement should be rephrased to better reflect that the model has already been developed and evaluated in this study rather than being an ongoing objective. The revised sentence will be as follows: “This paper presents an automated AI-enabled LUS model for real-time vessel segmentation, which has been developed to support liver surgery and improve intraoperative guidance.”
>
> 3. At the time of paper submission, the radiologist had not yet completed the final verification of remaining annotations, which is why we stated that they were “currently under final review”. Since then, the review has been completed, and all annotations have been confirmed as medically acceptable, with no further changes required. We will update the manuscript accordingly to reflect the final status.
>
> 4. All sonograms used in this study were extracted from retrospective laparoscopic ultrasound videos recorded using a commercial ultrasound device, therefore, raw data was not available. We agree that providing these details would clarify the data acquisition process, and we will include this information in the Dataset and Data Split subsection of the revised manuscript.
>
> 5. The ultrasound video frames contain various non-anatomical on-screen elements such as text overlays, depth and distance markers, and other interface graphics from the ultrasound device. If included during training, these elements could unintentionally provide the neural network with irrelevant features and lead it to associate them with anatomical structures, thus reducing the performance. To minimize this effect, we applied dynamic contour detection, followed by the remaining preprocessing steps to ensure that clinically relevant image content was used for training.  If space allows, such clarification will be added to the revised paper.
>
> 6. During training, we followed the common setup from the literature, where the model predicts the segmentation for the central frame using both past and future frames to learn temporal context. However, during real-time inference, the future frame is not yet available, so we used the two previous frames instead. This caused a small misalignment between training and inference. We acknowledge this limitation and plan to retrain the model using only past frames in future work to better match real-time conditions.
>
> 7. During the initial model development, we ran experiments with several common geometric and color-based augmentations. We observed that different encoder architectures responded inconsistently to specific augmentation combinations. For example, some benefited from horizontal flips, shifts, or brightness variations, others showed reduced performance under the same conditions. Despite the tuning, the results remained unstable across backbones. To ensure a fair comparison between models and to avoid biasing performance toward a particular encoder, it was decided to train all models without augmentation. If space allows, the choice of skipping augmentations will be clarified in the revised paper.
> 8. During the initial experiments, automatic mixed precision (AMP) training was applied to improve training speed and reduce GPU memory usage. We did not observe any noticeable drop in Dice scores when using AMP. However, we did not perform a dedicated experiment to measure its impact on Dice or training time. Since we lack exact comparative data, we agree that this statement should not mention that accuracy was not affected. We will revise the text to state that mixed precision was used to accelerate training and reduce memory requirements, without claiming that accuracy remained unchanged.
>
> 9. We agree that early stopping does not itself prevent overfitting but rather limits unnecessary training once the validation performance stops improving. We will revise the phrasing in the manuscript to accurately reflect on this.
>
> 10-13: We thank the reviewer for the comments and for raising a concern about how the dataset was handled. We acknowledge that our dataset is relatively small (11 total cases, 2,200 ultrasound images). The limited dataset size restricts the number of validation cases per fold. To address this, one full random case (Case 8) was excluded from cross-validation and reserved for final testing. The remaining 10 cases were used for 5-fold CV. In each fold, 8 cases (1,600 images) were used for the training set and 2 cases (400 images) for the validation set. Despite the small number of validation cases, the SD across folds was low, suggesting consistent model performance without strong fold-to-fold variation. We recognize that 5-fold CV is not ideal for very small datasets, but it remains a commonly used approach in medical imaging studies with limited data.
> For final model training, we applied a standard 90/10 split on the 10 training cases to maximize the amount of training data with only a small portion for validation. We finally evaluated the model on the earlier excluded test set of 200 images.
>
> We would also like to acknowledge another possible limitation  related to our dataset. The collected data consisted of 11 cases obtained from 9 patients with liver pathology. All images were manually annotated using locally developed annotation tool based on SAM 2 model, and verified by an experienced radiologist to ensure that they are clinically acceptable.  Patients 2-8 each have one acquisition of 200 images. Patient 1 had three separate laparoscopic ultrasound acquisitions (200 images each) , at different occasions (data, time) and representing different anatomical views. Each patient 1 acquisition captured different liver segments and different viewing angles, which was also confirmed by the experienced radiologist. We treated them as independent cases to make full use of the available annotated data. This was a pragmatic choice given the limited dataset size, to maximize and utilize all available data. We also verified that Case 8, which was randomly selected for independent testing, did not belong to the three acquisitions from Patient 1.
> If accepted, we will add this clarification in the revised manuscript under the Dataset and Data Split subsection and include additional illustrative images in the Appendix. We are also willing to share the dataset with interested researchers under an institutional data - sharing agreement, as we recognize the rarity and potential value of this dataset for the community.
>
> 14. We agree that the current color choices in Figures 2 and 3 are not the most optimal, in the revised version, we will adjust the overlay colors to use brighter and higher-contrast tones to better highlight the relevant regions and improve visual clarity.
>
> We thank the reviewer again for recognizing the value of this work.

---

### Official Review · Reviewer_U6LV · 2025-10-06
**AI-based temporal segmentation of vessels in laparoscopic ultrasound**

**Rating:** 4
**Confidence:** 4

**Summary:**

The study presents AI-based temporal segmentation of vessels in laparoscopic ultrasound using U-Net based encoder-decoder with ResNet backbone.  The solution use triplets of consecutive frames to leverage temporal information of sequential ultrasound data that were privately acquired for research purpose. The authors investigates multiple architectures in order to specify the best one.

**Strengths:**

- usage of real-world acquired LUS liver video data (is the video going to be publicly available upon acceptance? )
- multiple architectures evaluated
- description of metrics such as FPS, model size and Time for different encoders
- well-written and coherent

**Weaknesses:**

- missing citations for model architectures
- authors claims to use multiple loss functions but no results were included to confirm why the BCE was chosen as the final one,
- the evaluation is done only on one private dataset, the study would benefit from evaluation on available public videos,

**Justification:**

Generally well-written and motivated paper. It would be beneficial for the community If authors share the video and code for further reproducibility of the results.

---

> ### Author Rebuttal · Authors · 2025-10-22
>
> To Reviewer U6LV
>
> Author’s Response:
> We sincerely thank the reviewer for highlighting the Strengths and Weaknesses of the work.
>
> Response to the question raised in Strength’s part:
> We are willing to share the dataset with interested researchers under an institutional data-sharing agreement, as we recognize the rarity and potential value of this dataset for the community.
>
> Response to the 1st weakness:
> In the revised version, we will include the missing citations for model architecture.
>
> Response to the 2nd weakness:
> If space allows, we will include results from initial experiments with different loss functions made on Vanilla UNet, otherwise we will add it to the Appendix section.
>
> Response to the 3rd weakness:
> To our knowledge, there is no publicly available liver laparoscopic ultrasound dataset with pixel level vessel annotations.
>
> We thank the reviewer again for recognizing the value of this work and its contribution to AI-based liver vessels segmentation using ultrasound images.

---

### Official Review · Reviewer_sg7C · 2025-10-08
**A decent description of an analysis pipeline**

**Rating:** 4
**Confidence:** 4

**Summary:**

The paper addresses the problem of segmenting vessels in laparoscopic images. As such, the work is well motivated. The authors choose to use several variants of U-Nets, which is also reasonable. They apply pre-processing and enhance the model with feature extraction and modified input layers. The results are as expected.

**Strengths:**

The work is well-motivated, and the paper is well-structured and clear. Several variants of segmentation networks have been tested. It is nice to see the experiments on the triple setup.

**Weaknesses:**

The work is primarily confirmatory: the tested architectures and observed performance trends are consistent with existing understanding of UNet-based segmentation. The contribution lies more in documentation of results than in generating new insights. The authors could have emphasized what these results teach us about model design and dataset characteristics. For example documenting the effect of pre-processing, testing how the triple setup performs if the three slices are further apart.

**Justification:**

The paper is a well-motivated and clear documentation of solid work.

---

> ### Author Rebuttal · Authors · 2025-10-22
>
> To Reviewer sg7C
>
> Author’s Response: We sincerely thank the reviewer for the thoughtful comments. We acknowledge the reviewer’s observation and in the revised version, we will expand the discussion section to highlight the effect of pre-processing steps (e.g., CLAHE, resizing with aspect ratio) on segmentation accuracy. Additionally, if space allows, we will include a brief analysis exploring how varying the triple slice setup influences performance, e.g., neighboring frames vs three slices further apart vs 5 slices.
>
> We thank the reviewer again for recognizing the value of this work and its contribution to AI-based liver vessels segmentation using ultrasound images.

---

### Official Review · Reviewer_8EyZ · 2025-10-08
**A promising start with method lacking innovation and justification**

**Rating:** 4
**Confidence:** 4

**Summary:**

The paper proposes a real-time  laparoscopic ultrasound (LUS) image segmentation method. The authors seem to introduce five pre-processing steps on top of well known U-Net and ResNet18 neural network architectures. The preprocessing steps help to reduce noise in the LUS images and overcome the unclear vessel boundaries inherent in LUS. Also a temporal triplet input process is used to enhance realtime processing and accuracy of segmentation. The authors evaluate multiple encoder models including ResNet18, ResNet50,  MobileNet v2, Vanilla U-Net, DenseNet121 and IncResNetV2. The IncResNetV2 model achieved the best performance in their evaluation beating the ResNet18 by a very small margin.

Low availability of real-world data is a barrier to research in sophisticated domains like the LUS images used in the paper. To ensure robustness and generalisation, the researchers locally acquired a fully anonymised LUS liver video dataset comprising 2,200 frames from 11 separate cases. Developing a generalised model using 11 use cases seems like a ambitious attempt. According to the authors, the 2200 frames are manually annotated - which is an appreciable effort. Having said that, the 2200 frames correspond to 11 cases i.e., there is not enough variations in the 2200 images.

The authors demonstrated that the temporal triplet input (processing three consecutive frames instead of a single frame) provides stability in the segmented images. Feeding previous and next frames of the target frame provided 0.4% accuracy boost. This claim regarding triplet setup is interesting but lacking justification and sound evaluation.

Suitability to real-time deployment is one of the features evaluated in the paper. The finding is the most promising model in Table 1 i.e., IncResNetV2 is not suitable for deployment. According to Table 3, IncResNetV2 model achieves a FPS of 21 whereas according to the paper 30 FPS is required for realtime deployment.

**Strengths:**

The paper present a system for realtime image segmentation of Laparoscopic Ultrasound Images. Strengths of the paper:

1. The paper addresses an important challenge in LUS that reflects real-world constraints: noisy images from ultrasound scans.

1. The proposed approach is directly relevant to healthcare screening that has a high social value.

1. The problem formulation and choice of preprocessing steps, model architecture etc are evaluates with the available real world data.

**Weaknesses:**

The paper starts well in the introduction but diverge on page 2. There are major concerns regarding the innovation, justification of the method, evaluation and evaluation of related works.

1. **Innovation**: the paper does not clearly state where the innovation of the proposed approach shines. Without clear statements on the differentiating factors in the proposed approach, it reads like a class project rather than an academic reaserach.
1. **justification**: None of the used preprocessing steps are justified using relevant publication or empirical evidence. For example, why only triplet? why not more than 3 frames?
1. **evaluation**:  there is a serious concern regarding evaluation setup. There are 11 use cases. A 90/10 split will provide 9 or 10 training cases and 1 or 2 validation cases. Considering 10 cases in the training set, the 5 fold cv includes 2 cases per fold i.e., cv performance is measured on two cases in each iteration.
1. ** evaluation of related works**: the paper lacks a comprehensive evaluation of exisiting research in realtime image segmentation methods.

**Justification:**

**Limitations regarding data quantity and clinical verification**
A critical weakness of the paper is the scale and rigour of the clinical data used in training and evaluation:
- The model was trained and evaluated using LU videos from only 11 instances (that may come from 11 different patients or less that 11 patients if multiple cases correspond to the same patient). The leave-one-case-out evaluation attempts to maximise generalisation but the small sample size may raise concerns about the diversity and generalisability of the model.
- The first contribution (line 120, page 2) explicitly notes a lack of complete final clinical verification of the ground truth data. This suggests that the fundamental evaluation results in the paper might not yet be fully verified or clinically approved, introducing uncertainty in the reported "correctness" of the results.

**Lack of novelty**
- The core architectural components of the presented model rely entirely on existing, established concepts, which might be perceived as lacking significant novelty of the paper.
- The claimed key enhancement for stability, the triplet input setup (using sequential frames), is similar to setups already used for LUS-CT co-registration and for object recognition. The paper's contribution is the adaptation of this method to LUS rather than the creation of a fundamentally new architectural component or addressing a well justified problem in this domain.

**Others**:
- The final selected model ResNet18 intentionally sacrificed the model achieved the highest demonstrated segmentation accuracy (IncResNetV2), which could be viewed as a limitation without further justification.
- The slower throughput InceptionResNetV2 model achieved the highest Dice score (0.888) and the highest recall (0.867). The choice to deploy ResNet18 (Dice 0.879) was based on efficiency and real-time capability, potentially indicating a lack of thorough exploration of "how to achieve the highest accuracy of InceptionResNetV2 while maintaining real-time speed (e.g., through further optimisation of layers and/or connection in the network)".

---

> ### Author Rebuttal · Authors · 2025-10-22
>
> To Reviewer 8EyZ
>
> Author’s Response:
>
> We sincerely thank the reviewer for highlighting the Strengths and Weaknesses of the work.
>
> Response to 1st weakness:
> We acknowledge the reviewer’s observation and will make the innovation statement clearer at the end of the Introduction section in the revised version of the manuscript.
>
> Response to 2nd weakness:
> In the revised manuscript, we will provide relevant scientific references supporting each preprocessing component, except for the dynamic contour detection, which was developed locally as a novel, task-specific solution.
> We acknowledge that a systematic comparison of different temporal window sizes was not performed in this study and was selected based on the literature. If space allows, we will include a brief analysis exploring how varying selected slice setup influences performance, e.g., neighboring three frames vs three slices further apart vs 5 frames, and run the final model with these setups to compare.
>
> Response to 3rd weakness:
> We acknowledge that our dataset is relatively small (11 total cases, 2,200 ultrasound images), but it is comparable to other studies in liver vessel segmentation [1],[2]. The limited dataset size restricts the number of validation cases per fold. To address this, one random case (Case 8) was excluded from cross-validation and reserved for final testing. The remaining 10 cases were used for 5-fold CV. In each fold, 8 cases (1,600 images) were used for the training set and 2 cases (400 images) for the validation set. Despite the small number of validation cases, the standard deviation across folds was low, suggesting consistent model performance without strong fold-to-fold variation. We recognize that a 5-fold CV is not ideal for very small datasets, but it remains a commonly used approach in medical imaging studies with limited data [1].
> For final model training, we applied a standard 90/10 split on the 10 training cases to maximize the amount of training data but also kept a small portion for internal validation. We finally evaluated the model on the earlier excluded test set of 200 images.
> If approved, we will clearly note this limitation as a part of discussion and mention possible future improvements, such as evaluating on smaller 3-fold CV or leave-one-out cross-validation (LOOCV), which could further strengthen the generalizability of the evaluation.
>
> We would also like to acknowledge another possible limitation related to our dataset. The collected data consisted of 11 cases obtained from 9 patients with liver pathology. All images were manually annotated and verified by an experienced radiologist to ensure that they are clinically acceptable.  Patients 2-8 each have one acquisition of 200 images. Patient 1 had three separate laparoscopic ultrasound acquisitions (200 images each), at different occasions (data, time) and representing different anatomical views. Each patient 1 acquisition captured different liver segments and different viewing angles, which was also confirmed by the experienced radiologist. We treated them as independent cases to make full use of the available annotated data. This was a pragmatic choice given the limited dataset size, to maximize and utilize all available data. We also verified that Case 8, which was randomly selected for independent testing, did not belong to the three acquisitions from patient 1.
>
> If accepted, we will add this clarification in the revised manuscript under the Dataset and Data Split subsection and include additional illustrative images in the Appendix. We are also willing to share the dataset with interested researchers under an institutional data-sharing agreement, as we recognize the rarity and potential value of this dataset for the community.
>
> [1] Tanaka, K., Kurihara, T., Takahashi, Y., Onogi, S., Sugino, T., Nakajima, Y., … & Masuda, K. (2024). Segmentation of Liver Blood Vessel in Ultrasound Images Using Mask R-CNN. Advanced Biomedical Engineering, 13, 379–388. DOI: 10.14326/abe.13.379
> [2] Ansari, M. Y., Alzahrani, M., Rizwan, M., & Khusro, S. (2023). Dense-PSP-UNet: A neural network for fast inference liver US segmentation. Computers & Electrical Engineering, 110, 108404. DOI: 10.1016/j.compeleceng.2023.108404
>
> Response to 4th weakness:
>
> We appreciate the reviewer’s comment and agree to update the part of the Introduction section to include a clearer comparison of existing methods in real-time image segmentation.
>
>
> We thank the reviewer again for recognizing the value of this work and its contribution to AI-based liver vessels segmentation using ultrasound images.

---

### Meta-Review · Area_Chair_y5RN · 2025-10-31

**Recommendation:** Accept (Poster)
**Confidence:** 3

**Metareview:**

The paper addresses an important clinical problem, it achieves practical results (real-time performance with good accuracy) and has thorough experimental comparison, and clear presentation. The lack of strong methodological novelty and dataset limitations presents a counterweight to this, but I believe the clinical relevance and solid execution justify acceptance, as is also the consensus among the reviewers.

---

### Decision · Program_Chairs · 2025-11-05

**Decision:**

Accept (Spotlight)

**Comment:**

We recommend an oral and a poster presentation given the AC and reviewers recommendations.

A spotlight presentation refers to a poster selected for an oral highlight but not designated as a full oral presentation per the AC’s recommendation.